# In vitro reconstitution of dynamically interacting integral membrane subunits of energy-coupling factor transporters

Inda Setyawati[1,2†], Weronika K Stanek[1†‡], Maria Majsnerowska[1†§], Lotteke J Y M Swier[1#], Els Pardon[3,4], Jan Steyaert[3,4], Albert Guskov[1,5*], Dirk J Slotboom[1*]

[1]Groningen Biomolecular Sciences and Biotechnology Institute, University of Groningen, Groningen, Netherlands; [2]Biochemistry Department, Bogor Agricultural University, Bogor, Indonesia; [3]Structural Biology Brussels, Vrije Universiteit Brussel, VUB, Brussels, Belgium; [4]VIB-VUB Center for Structural Biology, VIB, Brussels, Belgium; [5]Moscow Institute of Physics and Technology, Dolgoprudny, Russian Federation

**\*For correspondence:**
a.guskov@rug.nl (AG);
d.j.slotboom@rug.nl (DJS)

[†]These authors contributed equally to this work

**Present address:** [‡]Institute of Bioorganic Chemistry, Polish Academy of Sciences, Poznan, Poland; [§]QPS Netherlands B.V. Ingang, Groningen, Netherlands; [#]Department of Pathology and Medical Biology, University Medical Centre Groningen, Groningen, Netherlands

**Competing interests:** The authors declare that no competing interests exist.

**Abstract** Energy-coupling factor (ECF) transporters mediate import of micronutrients in prokaryotes. They consist of an integral membrane S-component (that binds substrate) and ECF module (that powers transport by ATP hydrolysis). It has been proposed that different S-components compete for docking onto the same ECF module, but a minimal liposome-reconstituted system, required to substantiate this idea, is lacking. Here, we co-reconstituted ECF transporters for folate (ECF-FolT2) and pantothenate (ECF-PanT) into proteoliposomes, and assayed for crosstalk during active transport. The kinetics of transport showed that exchange of S-components is part of the transport mechanism. Competition experiments suggest much slower substrate association with FolT2 than with PanT. Comparison of a crystal structure of ECF-PanT with previously determined structures of ECF-FolT2 revealed larger conformational changes upon binding of folate than pantothenate, which could explain the kinetic differences. Our work shows that a minimal in vitro system with two reconstituted transporters recapitulates intricate kinetics behaviour observed in vivo.

## Introduction

ATP-binding cassette (ABC) transporters are membrane protein complexes that mediate translocation of molecules across the bilayer fuelled by ATP hydrolysis. All ABC transporters contain two ATPase domains or subunits (also called Nucleotide Binding Domains, NBDs) that share highly conserved motifs, and two transmembrane subunits or domains (TMDs) (for reviews, see references *ter Beek et al., 2014*; *Thomas and Tampé, 2020*). In prokaryotes three major classes of ABC transporters involved in import have been distinguished based on the fold of the transmembrane domains. In two of these classes (named Type I and Type II ABC transporters) (*Thomas et al., 2020*), extracellular or periplasmic proteins are required for substrate binding and delivery to the transporter, called substrate-binding proteins or domains (SBPs or SBDs).

The third class (Type III ABC transporters) consists of energy-coupling factor (ECF) transporters. Like all ABC transporters, ECF transporters contain two cytosolic ATPases (often a heterodimer of EcfA and EcfA') and two transmembrane domains (*Rodionov et al., 2009*), the latter with unique architectures. In ECF-type ABC transporters, the two membrane subunits are not related in structure, and only one transmembrane protein (EcfT) interacts with the NBDs via two long coupling helices (*Erkens et al., 2011*; *Xu et al., 2013*; *Wang et al., 2013*; *Rempel et al., 2019*). The single

membrane subunit EcfT and the two ATPases together form the so-called energy-coupling factor or ECF module. In contrast, in Type I and Type II ABC transporters, the membrane subunits are homologous or identical and both contain a coupling helix to transmit conformational changes from the NBDs to the transmembrane part of transporter (*Hollenstein et al., 2007*; *Oldham et al., 2007*; *Locher et al., 2002*).

Substrate binding in ECF transporters is mediated by the second integral membrane subunit, termed S-component. Despite their highly diverse amino acid sequences, all structurally-characterised S-components share a conserved core fold of six α-helices (*Erkens et al., 2011*; *Berntsson et al., 2012*; *Zhang et al., 2010*). It has been proposed that the association of the S-component with and dissociation from the ECF module are steps of the transport mechanism (*Rodionov et al., 2009*; *Henderson et al., 1979a*; *Karpowich et al., 2015*). In the solitary state, the α-helices of S-components adopt a classical transmembrane orientation (approximately perpendicular to the membrane plane) (*Erkens et al., 2011*; *Faustino et al., 2020*), and the proteins invariably exhibit high affinity for their specific substrates, with low- to subnanomolar $K_D$ values (*Erkens et al., 2011*; *Rempel et al., 2019*; *Berntsson et al., 2012*; *Swier et al., 2016*; *Slotboom, 2014*; *Santos et al., 2018*; *Jochim et al., 2020*; *Erkens et al., 2012*). The high affinity allows them to scavenge scarce substrates from the environment. Association with the ECF complex has two consequences: First, the S-component topples over in the membrane, thereby moving the bound substrate across the membrane with some of the α-helical segments eventually orienting themselves parallel to the membrane plane (*Xu et al., 2013*; *Wang et al., 2013*; *Rempel et al., 2019*). Such a mechanism has been classified as 'elevator-type' (*Garaeva and Slotboom, 2020*; *Drew and Boudker, 2016*). Second, the substrate-binding site is disrupted, leading to facilitated release into the cytosol (*Swier et al., 2016*; *Santos et al., 2018*). While in 'group I' ECF transporters the ECF module is dedicated for interaction with a single S-component, in group II ECF transporters, the same ECF module can associate with different S-components, and hence be used to assist the transport of different substrates (*Rodionov et al., 2009*). Organisms using group II ECF transporters produce the S-components according to the need for a particular substrate, which forces them to compete for the limiting amount of ECF modules for transport (*Rodionov et al., 2009*; *Henderson et al., 1979b*; *Majsnerowska et al., 2015*).

The unusual transport mechanism of group II transporters with a shared ECF module was hypothesised first by *Henderson et al., 1979a*, who showed that transport of one vitamin in cells of *Lactobacillus casei* is inhibited by other vitamins with complex inhibition kinetics. More recently, investigations in a heterologous expression system revealed that indeed different S-components can compete for a shared ECF module in a substrate-concentration dependent manner (*Majsnerowska et al., 2015*). In detergent solution, biochemical and structural studies have provided insight in the shared use of the ECF module and revealed a common interaction surface of different S-components (*Santos et al., 2018*; *Zhang et al., 2014*; *Swier et al., 2016*). The available data indicate that association and dissociation of S-components take place as a part of transport cycle, but a minimal experimental in vitro system with multiple ECF transporters reconstituted in proteoliposomes, is necessary to substantiate the observations.

Here, we used two purified group II ECF transporters from *Lactobacillus delbrueckii* (the folate transporter ECF-FolT2 and the pantothenate transporter ECF-PanT), which we reconstituted in liposomes to test whether competition between S-components is intrinsic to the proteins, or that it might be dependent on unidentified interaction partners present in the cellular environment. We chose to perform these studies on purified proteins in proteoliposomes instead of detergent solution because the detergent micelle could affect S-component dissociation and association (*Karpowich et al., 2015*). We also decided not to use membrane-mimicking systems such as lipid nanodiscs for the experiments, because they suffer from a limited size of the bilayer, which prevents reconstitution of multiple proteins, and may affect their association dynamics (*Finkenwirth et al., 2017*). Furthermore, lack of compartmentalisation in detergent or nanodiscs environment makes it impossible to assay vectorial transport. Using the proteoliposomes system we show that the subunit composition of ECF transporters is dynamic, with the S-components FolT2 and PanT associating with, and dissociating from the same ECF module. Moreover, while these integral membrane proteins compete for the same interaction site on the ECF module, the kinetics of competition differs for the two substrates. We conclude that dissociation of the S-component from ECF- transporter complex and subsequent association of the same or different S-component is a part of transport

cycle. Because of the recurring association and dissociation of S-components, ECF transporters are a potential model for studying more general properties of membrane protein interactions in the lipid bilayers.

## Results

The Gram-positive bacterium *L. delbrueckii* contains eight different S-components that make use of the same ECF module (*Rodionov et al., 2009*; *Swier et al., 2016*; *Overbeek et al., 2005*). To study the association with and dissociation from the ECF module we selected two S-components, namely FolT2, which is specific for folate and PanT, which is predicted to bind pantothenate. We overproduced and purified the complete complexes ECF-FolT2 and ECF-PanT each containing four subunits. In addition, we were able to purify large amounts of the solitary S-component FolT2 (in the absence of the ECF module) in a stable state (*Swier et al., 2016*). Solitary PanT was marginally stable in detergent solution in all tested conditions, and only small quantities of purified protein could be produced. Therefore, we designed our experiments in such a way that PanT was always purified in complex with the ECF module, but was allowed to dissociate from ECF module once reconstituted in the liposomes (see below). Only for a few crucial control experiments we used purified, solitary PanT.

### PanT and FolT2 form a functional transport complex with the same ECF module

The purified ECF-transporter complexes ECF-FolT2 and ECF-PanT were both active when reconstituted into proteoliposomes (*Figure 1*), and mediated ATP-dependent uptake of folate and pantothenate, respectively. The accumulation of radiolabelled substrate in the proteoliposomes' lumen was strictly dependent on the presence of lumenal $Mg^{2+}$-ATP (*Figure 1*). Transport of folate and pantothenate was possible only in the presence of a dedicated S-component. We could not detect transport of pantothenate by ECF-FolT2 or folate transport by ECF-PanT (*Figure 1*), confirming that the substrate specificity of the ECF transporters is determined entirely by the specific S-components (*Figure 1*).

We determined the apparent $K_m$ for substrate transport by measuring initial rates of uptake of the vitamin substrate into proteoliposomes, at a fixed ATP concentration. The apparent $K_m$ values for pantothenate and folate transport were in the nanomolar range (*Figure 2ab*), consistent with the notion that ECF transporters are high-affinity scavengers of micronutrients (*Rodionov et al., 2009*; *Rempel et al., 2019*; *Duurkens et al., 2007*; *Erkens and Slotboom, 2010*). The dependence of the transport rates on the vitamin concentrations were hyperbolic, both for pantothenate and folate transport, indicating that there was no cooperativity, consistent with a single binding site in the S-component (*Erkens et al., 2011*; *Zhang et al., 2010*; *Swier et al., 2016*; *Zhao et al., 2015*). We also determined the dependence of the transport rates on the concentration of ATP, and found $K_m$ values in the mM range (*Figure 2cd*), in line with $K_m$ values reported for other ABC transporters. The dependence of the transport rates on the ATP concentration was sigmoidal, showing that there is cooperativity between the ATP binding sites. It is noteworthy that previous assays for ATPase instead of transport activity revealed hyperbolic relations between the ATP concentration and the hydrolysis rate (*Karpowich et al., 2015*). This observation underlines that care needs to be taken when using ATPase assays to obtain insight in the transport mechanism. While $K_m$ values and Hill coefficients were very similar for folate and pantothenate transport, the apparent maximal rates of transport differed somewhat for the two substrates, which may suggest intrinsic differences between transport mediated by FolT2 and PanT, but it must be noted that unequal activity losses during purification and reconstitution could also account for these differences. Moreover, both protein complexes can orient either in the right-side-out or inside-out orientation in the liposomal membrane (*Swier et al., 2016*). Therefore, the apparent $V_{max}$ values are likely underestimations. In contrast, a mixed orientation does not affect the $K_m$ because the use of different chemical compositions in the lumenal and external solutions (ATP, transported substrate) allowed us to probe the uptake activity of proteins in the right-side-out orientation, with the proteins in the other orientation remaining 'invisible'.

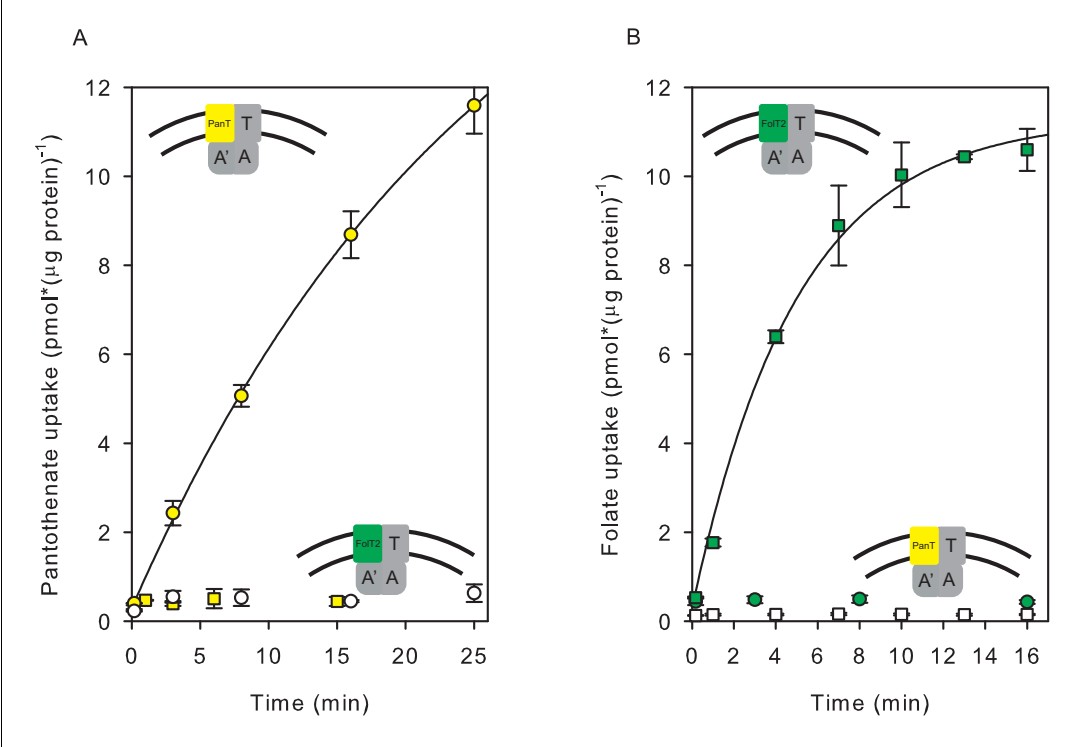

**Figure 1.** Transport of [³H]pantothenate and [³H]folate into proteoliposomes. (**A**) Yellow and white circles: Pantothenate uptake by ECF-PanT into proteoliposomes containing 10 mM $Mg^{2+}$-ATP or $Mg^{2+}$-ADP in the lumen, respectively; Yellow squares: Pantothenate uptake by ECF-FolT2 into proteoliposomes containing 10 mM Mg-ATP in the lumen. (**B**) Green and white squares: Folate uptake by ECF-FolT2 into proteoliposomes containing 10 mM $Mg^{2+}$-ATP or $Mg^{2+}$-ADP in the lumen, respectively; Green circles: Folate uptake by ECF-PanT into proteoliposomes containing 10 mM $Mg^{2+}$-ATP in the lumen. Error bars indicate standard deviation of triplicate measurements. The insets show schematic representations of the reconstituted systems used. FolT2 and PanT are coloured in green and yellow, respectively. The shared ECF module is shown in grey. The membrane boundaries are indicated by the two black lines. In *Figures 3–5* we use similar cartoons, and yellow and green symbols indicating pantotenate and folate uptake, respectively, with circles and squares indicating that the uptake was mediated by ECF-PanT and ECF-FolT2, respectively.

The online version of this article includes the following source data for figure 1:

**Source data 1.** Scintillation counts and analysis.
**Source data 2.** Scintillation counts and analysis.
**Source data 3.** Scintillation counts and analysis.
**Source data 4.** Scintillation counts and analysis.

## Exchange of S-components

We performed two types of experiments to show that the subunit composition of ECF transporters in liposomal membranes is dynamic. First, we co-reconstituted in proteoliposomes the whole complex ECF-PanT together with separately purified solitary FolT2. On average, one full ECF-PanT transporter was present per liposome together with 26 molecules of FolT2 (see below for calculation). The proteoliposomes with co-reconstituted ECF-PanT and solitary FolT2 showed transport activity for both folate and pantothenate (*Figure 3a*). Because solitary FolT2 cannot transport folate alone (*Figure 3a*) the observed uptake of radiolabelled folate in the liposomes shows that FolT2 had associated with the ECF module from the ECF-PanT complex.

We could also show that exchange happened in an experiment where ECF-FolT2 was reconstituted as a full complex together with solitary PanT, even though the yield and stability of the solitary PanT protein were low. These proteoliposomes were able to take up pantothenate (*Figure 3b*). The uptake of pantothenate was observed only when ECF-FolT2 was co-reconstituted, and not observed when the same amount of solitary PanT was reconstituted alone (*Figure 3b*).

In the second approach to demonstrate exchange of S-components in liposomes, we used a liposome system with two co-reconstituted full complexes: the wild type transporter complex for one

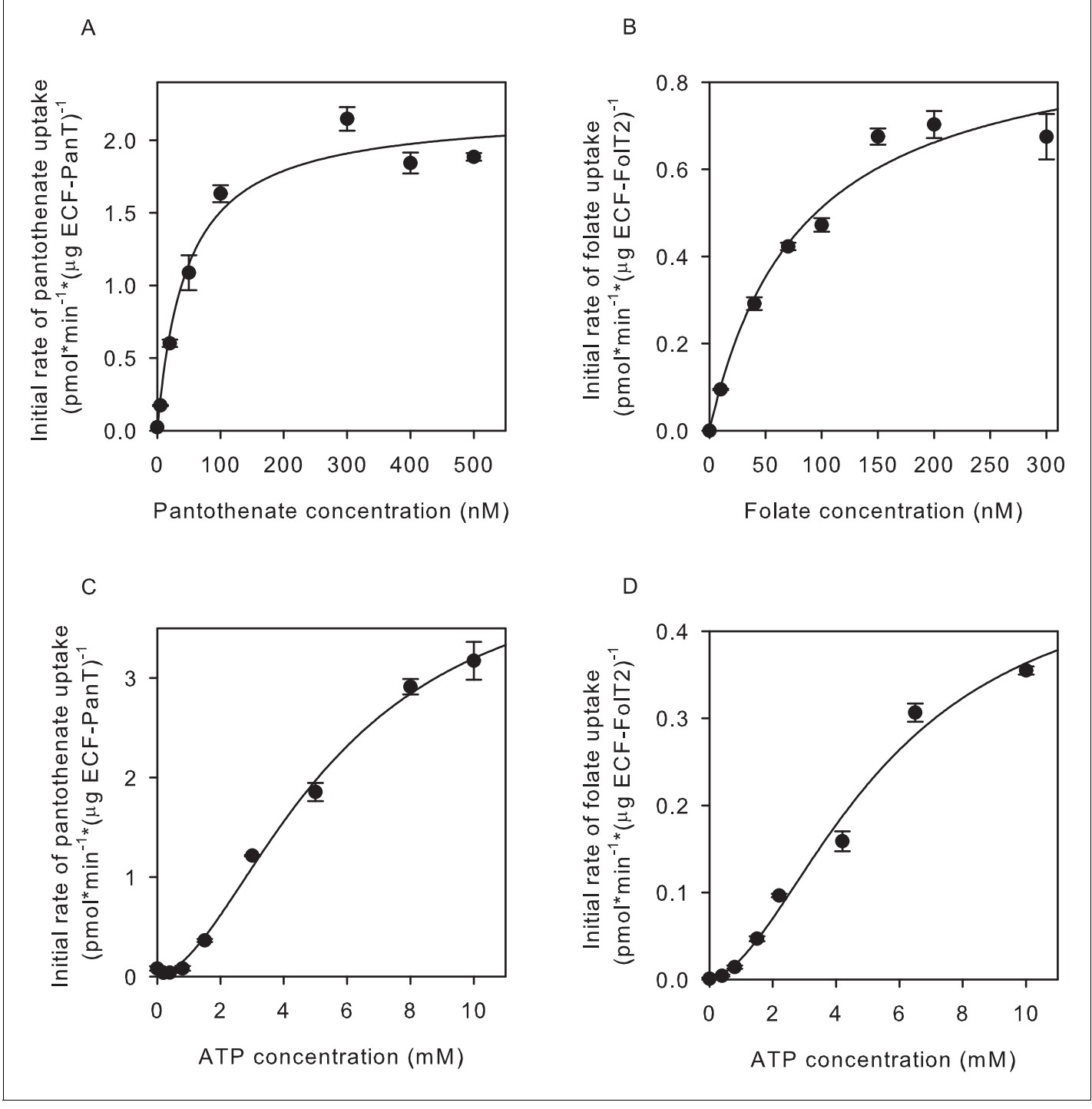

**Figure 2.** Determination of apparent $K_m$ and $V_{max}$ values for pantothenate and folate transport. (**A,C**) Initial rates of pantothenate transport by ECF-PanT into proteoliposomes as function of the pantothenate concentration (panel A, $Mg^{2+}$-ATP concentration 5 mM) or the ATP concentration (panel C, pantothenate concentration 100 nM). The apparent $K_m$ and $V_{max}$ values in the pantothenate-dependent measurements are 46 ± 11 nM and 2.2 ± 0.12 nmol/mg/min, respectively. For the ATP-dependent measurements 5.6 ± 1.0 mM and 4.4 ± 0.5 nmol/mg/min, respectively. (**B,D**) Initial rates of folate transport by ECF-FolT2 into proteoliposomes as function of the folate concentration (panel B, $Mg^{2+}$-ATP concentration 10 mM) or the ATP concentration (panel D, folate concentration 100 nM). The apparent $K_m$ and $V_{max}$ values in the folate-dependent measurements are 82 ± 20 nM and 0.93 ± 0.1 nmol/mg/min, respectively. For the ATP- dependent measurements 5.6 ± 1.7 mM and 0.5 ± 0.1 nmol/mg/min, respectively. Error bars indicate standard deviation of triplicate measurements.

The online version of this article includes the following source data for figure 2:

**Source data 1.** Scintillation counts and analysis.

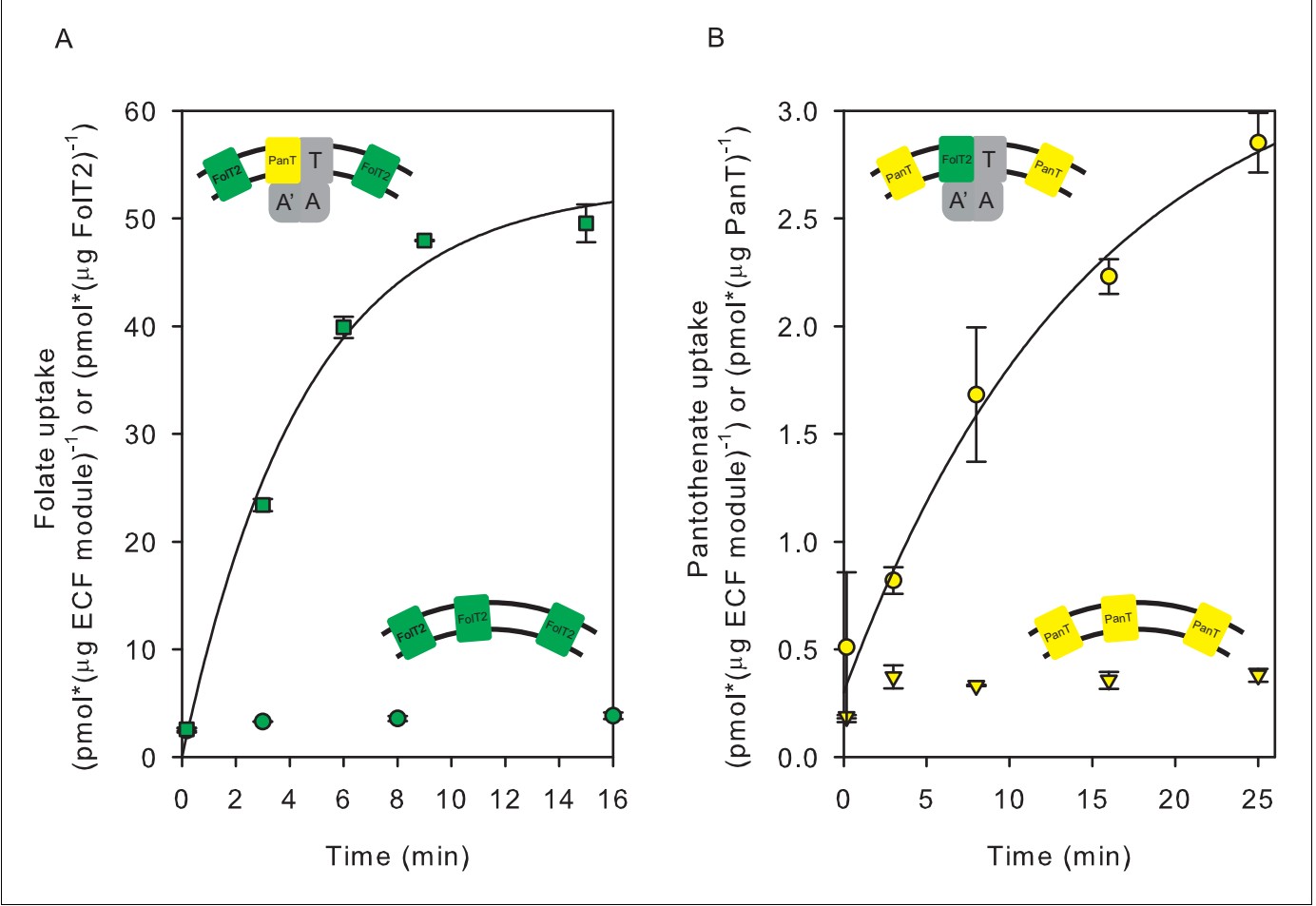

**Figure 3.** Exchanges of S-components in proteoliposomes reconstituted with complete and incomplete transporters. (**A**) Folate uptake into proteoliposomes reconstituted with FolT2 alone (green circles), or FolT2 in combination with ECF-PanT (green squares, molar ratio 26:1). (**B**) Pantothenate uptake into proteoliposomes reconstituted with PanT alone (yellow inverted triangles), or PanT in combination with ECF-FolT2 (Yellow circles). Since PanT was not very stable in detergent solution, the exact molar ratio in the combined reconstitution is unknown but likely to be much lower than in the experiment presented in panel A. The low amount of PanT could explain the reduced uptake rate. In all cases, 10 mM $Mg^{2+}$-ATP was present in the lumen. Error bars indicate standard deviation of triplicate measurements, apart from panel B, where the experiment was done in duplicate.

The online version of this article includes the following source data and figure supplement(s) for figure 3:

**Source data 1.** Scintillation counts and analysis.

**Figure supplement 1.** Pantothenate uptake into proteoliposomes reconstituted with FolT2 in combination with ECF-PanT.

---

substrate, and a complex with inactivated ATPase subunits for the other substrate. To inactivate the ATPases, we created a double mutant, in which the conserved catalytic glutamate residues in the Walker B motifs of both ATPase subunits (EcfA and EcfA') were changed into glutamine (E169Q in EcfA and E171Q in EcfA'). The glutamates are necessary to coordinate a water molecule for a nucleophilic attack on the bond between the γ- and ß-phosphate of ATP (*Oldham et al., 2007*; *Hanekop et al., 2006*). Therefore, glutamate-to-glutamine substitutions (EQ) are expected to be able to bind, but not hydrolyse ATP (*Oldham et al., 2007*; *Hanekop et al., 2006*). Folate and pantothenate transport activities of the double mutants were indeed at the level of background (*Figure 4*).

To test for S-component exchange, wild type and mutant ECF modules in complex with different S-components (FolT2 or PanT) were separately expressed and purified, and subsequently co-reconstituted into liposomes. Co-reconstitution of the active ECF-PanT complex with the mutated and inactive ECF-FolT2 complex resulted in transport of both substrates. The same result was obtained

eLife Research article

Biochemistry and Chemical Biology | Structural Biology and Molecular Biophysics

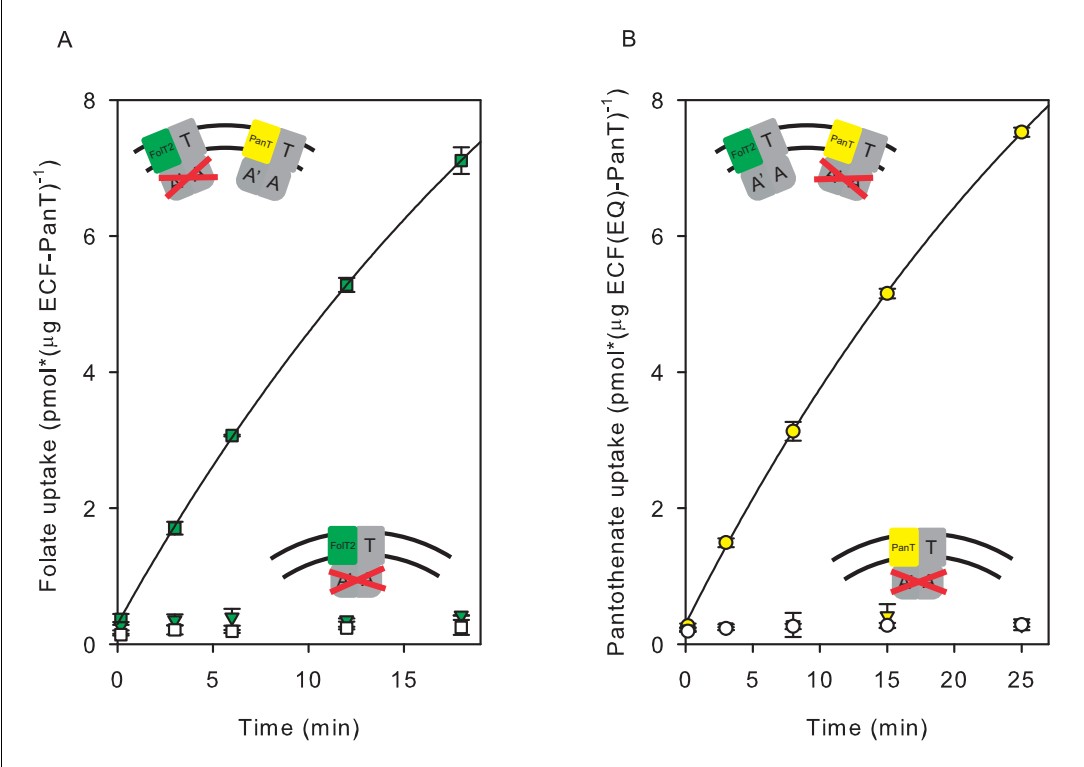

**Figure 4.** Transport of pantothenate and folate into proteoliposomes co-reconstituted with two full complexes one of which contained an ECF module with E-to-Q mutations in the Walker B motifs. In the cartoon insets, red crosses indicate the mutated ECF modules. (**A**) Folate uptake into proteoliposomes co-reconstituted with ECF-PanT and ECF(E-to-Q)-FolT2, containing 10 mM $Mg^{2+}$-ATP (green squares) or $Mg^{2+}$-ADP in the lumen (white squares), respectively; Green inverted triangles: Folate uptake into proteoliposomes reconstituted with only ECF(E-to-Q)-FolT2, containing 10 mM $Mg^{2+}$-ATP in the lumen, respectively;. (**B**) Pantothenate uptake into proteoliposomes co-reconstituted with ECF-FolT2 and ECF(E-to-Q)-PanT, containing 10 mM $Mg^{2+}$-ATP (yellow circles) or $Mg^{2+}$-ADP (white circles) in the lumen, respectively; Yellow inverted triangles: Pantothenate uptake by into proteoliposomes reconstituted with only ECF(E-to-Q)-PanT, containing 10 mM $Mg^{2+}$-ATP in the lumen. Error bars indicate standard deviation of triplicate measurements.

The online version of this article includes the following source data and figure supplement(s) for figure 4:

**Source data 1.** Scintillation counts and analysis.

**Figure supplement 1.** Control experiments for the ones shown in *Figure 4*, now with the two substrates swapped.

when active ECF-FolT2 was co-reconstituted with mutated ECF-PanT. These results show that S-component exchange had happened (*Figure 4* and *Figure 4—figure supplement 1*).

## Competition for a shared ECF module

In order to assay for competition of the two different S-components for association with the ECF module, we used the proteoliposomes with co-reconstituted ECF-PanT and solitary FolT2, as described above. To study competition of the S-components PanT and FolT2 for the same ECF module, the amount of the latter had to be limiting in the transport assays (thus mimicking the in vivo situation *Henderson et al., 1979a*), and therefore we reconstituted an excess of S-components relative to the ECF module in the liposomes. For a full ECF-transporter complex (Mw ~120 kDa), reconstitution using a protein-to-lipid ratio of 1:1000 (w/w) was expected to yield on average a single protein complex in each liposome of 400 nm diameter. Reconstitution of the solitary S-component FolT2 (Mw ~21 kDa) at a protein:lipid ratio of 1:250 (w/w), was expected to yield 26 protein molecules in each liposome. As discussed above, the proteins may orient either in the right-side-out or inside-out orientation in the liposomal membrane, but only transport by the right-side-out orientated proteins was assayed for, because we included ATP in the lumen, and added the transported substrates on the outside.

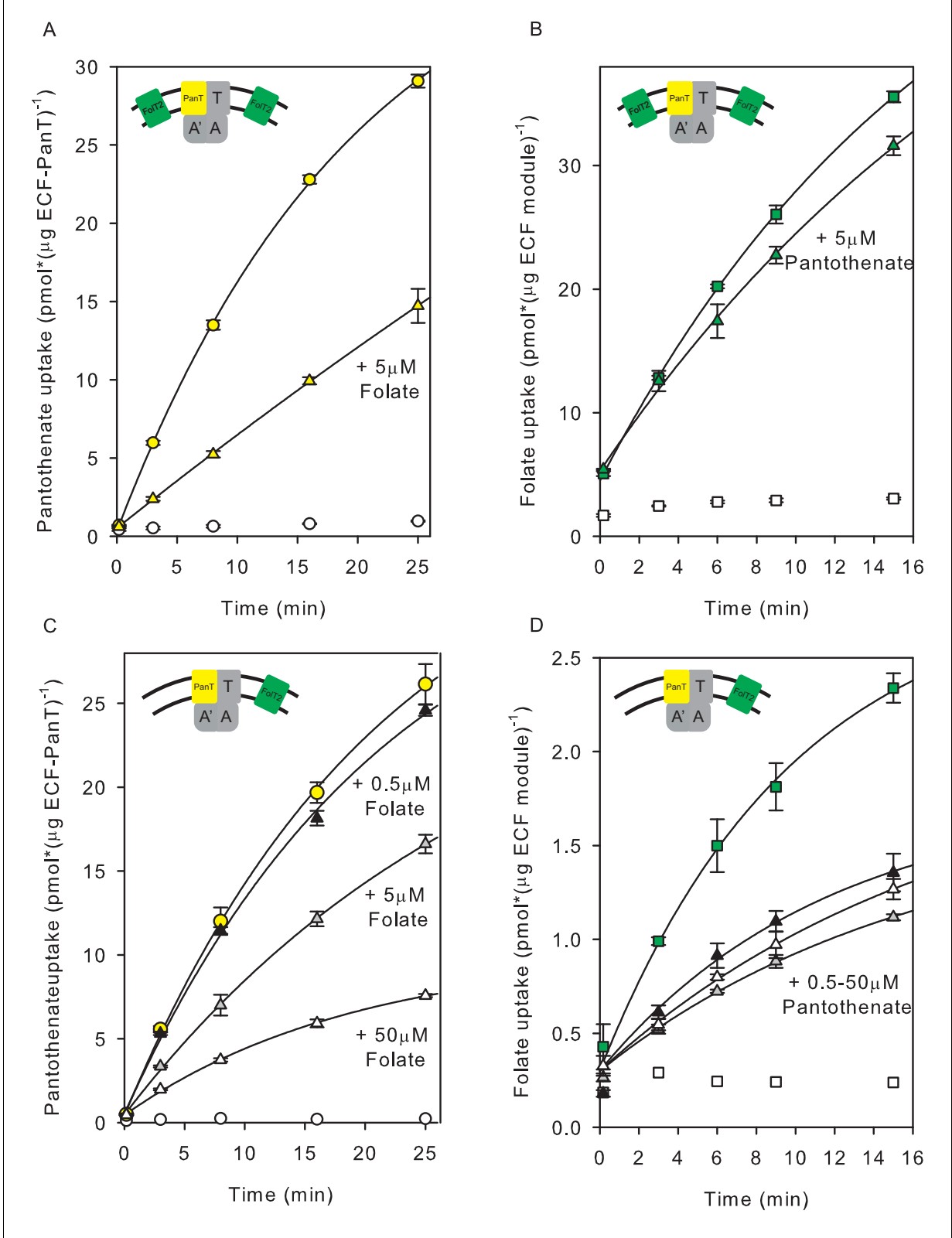

**Figure 5.** Inhibition of pantothenate uptake by folate and *vice versa* in proteoliposomes co-reconstituted with ECF-PanT and FolT2. (**A, B**) Uptake of radiolabelled pantothenate (**A**) and folate (**B**) into proteoliposomes co-reconstituted with ECF-PanT in protein-to-lipid ratio 1:1000 (w:w) and solitary FolT2 in ratio 1:250 (w:w), and loaded with 10 mM $Mg^{2+}$-ATP (coloured symbols) or $Mg^{2+}$-ADP (white symbols). Triangles: same as the conditions used for the black circles, but in the presence of 5 μM unlabelled folate (panel A) or pantothenate (panel B) as competing substrate. (**C, D**) same as panels A,

*Figure 5 continued on next page*

Figure 5 continued

B, but with reduced amount of FolT2 reconstituted (protein-to-lipid ratio 1:1000 (w:w) for both solitary FolT2 and ECF-PanT). The competing substrates were added at three different concentrations: 50 µM (white triangles), 5 µM (grey triangles) and 0.5 µM (black triangles). Error bars indicate standard deviation of triplicate measurements.

The online version of this article includes the following source data and figure supplement(s) for figure 5:

**Source data 1.** Scintillation counts and analysis.
**Figure supplement 1.** Lack of inhibition of radiolabelled pantothenate uptake by unlabelled folate in proteoliposomes containing only ECF-PanT.

Radiolabelled pantothenate was taken up readily into the proteoliposomes containing ECF-PanT co-reconstituted with FolT2 (*Figure 5a*, see also *Figure 3—figure supplement 1*). Addition of 5 µM unlabelled folate revealed a reduction of [³H]pantothenate uptake (*Figure 5a*). It therefore appears that the substrate-loaded S-component FolT2 competes more effectively for association with the ECF module than the apo-protein, in line with previous in vivo experiments (*Henderson et al., 1979a*; *Majsnerowska et al., 2015*). As a control, we also tested the effect of folate on the transport of pantothenate when only ECF-PanT had been reconstituted (*Figure 5—figure supplement 1*). As expected, in the absence of FolT2, folate did not affect pantothenate uptake by ECF-PanT.

Using the same preparation of proteoliposomes containing co-reconstituted ECF-PanT and solitary FolT2, we also followed the transport of radiolabelled folate. Folate transport (*Figure 5b*) was inhibited only slightly upon addition of unlabelled pantothenate (*Figure 5b*). The difference in sensitivity for the competing substrate in the panthothenate and folate transport assays could be a reflection of the considerable excess of FolT2 over PanT in the liposomes, as in each liposome on average one PanT molecule, and 26 molecules of FolT2 were present. To test whether the FolT2 excess could indeed explain the decreased sensitivity for added pantothenate, we reduced the amount of co-reconstituted FolT2 by fourfold to approximately seven FolT2 molecules per liposome. With the reduced amount of FolT2 in the liposomes, the inhibitory effect of pantothenate on folate uptake was indeed more pronounced (*Figure 5d*), showing that not only folate but also pantothenate enhances competition for the shared ECF module. However, there was a prominent difference in the dose-dependence of the effect of the competing substrate. The transport rate of radiolabelled pantothenate (*Figure 5c*) was inhibited by folate with strong dependence on the concentration in the range of 0.5 to 50 µM folate. In contrast, in the same concentration range there was no significant dose-dependence of the inhibitory effect of pantothenate on folate transport (*Figure 5d*).

## Crystal structure of ECF-PanT

The differences in dose-dependence of transport inhibition by folate and pantothenate (*Figure 5cd*) are remarkable, because ECF-FolT2 and ECF-PanT make use of identical ECF modules, and only differ in the S-components. Comparison of the structures of the two transporter complexes might provide insight in the structural basis of the kinetic differences. While crystal structures of ECF-FolT2 from *L. delbrueckii* were determined previously (*Swier et al., 2016*), structural information on ECF-PanT from the same organism is lacking. A structure of ECF-PanT form *L. brevis* is known (*Zhang et al., 2014*), but the PanT protein from this organism shares only 36% sequence identity with the one from *L. delbrueckii*, and thus may not be a suitable model. Therefore, we set out to determine a crystal structure of ECF-PanT from *L. delbrueckii*, but despite extensive trials, suitable crystals were not found. To overcome this problem, we generated nanobodies against ECF-PanT that could be used as a crystallisation chaperone (*Pardon et al., 2014*). One of the selected nanobodies (Nb81) bound with high affinity to the ECF module, and the ECF-PanT-Nb81 complex formed well-diffracting crystals. We solved a crystal structure of the complex at a resolution of 2.8 Å, the highest resolution for any ECF-transporter structure to date (*Figure 6a*, and *Table 1*). In the crystals, the asymmetric unit contains two copies of the ECF-PanT-Nb81 complex. The nanobodies are positioned centrally in the asymmetric unit, making extensive contacts with both copies of ECF-PanT in the unit, but do not participate in crystal contacts with proteins in neighbouring asymmetric units (*Figure 6a* and *Figure 6—figure supplement 1*). Consistently, the nanobodies cause dimerisation of the ECF-PanT complex in detergent solution (*Figure 6*, *Figure 6—figure supplement 2*). An elaborate network of hydrogen bonds, electrostatic interactions, cation-π and π-π interactions between the nanobody and EcfA and EcfA' seems to stabilise the protein in a single conformation, which may

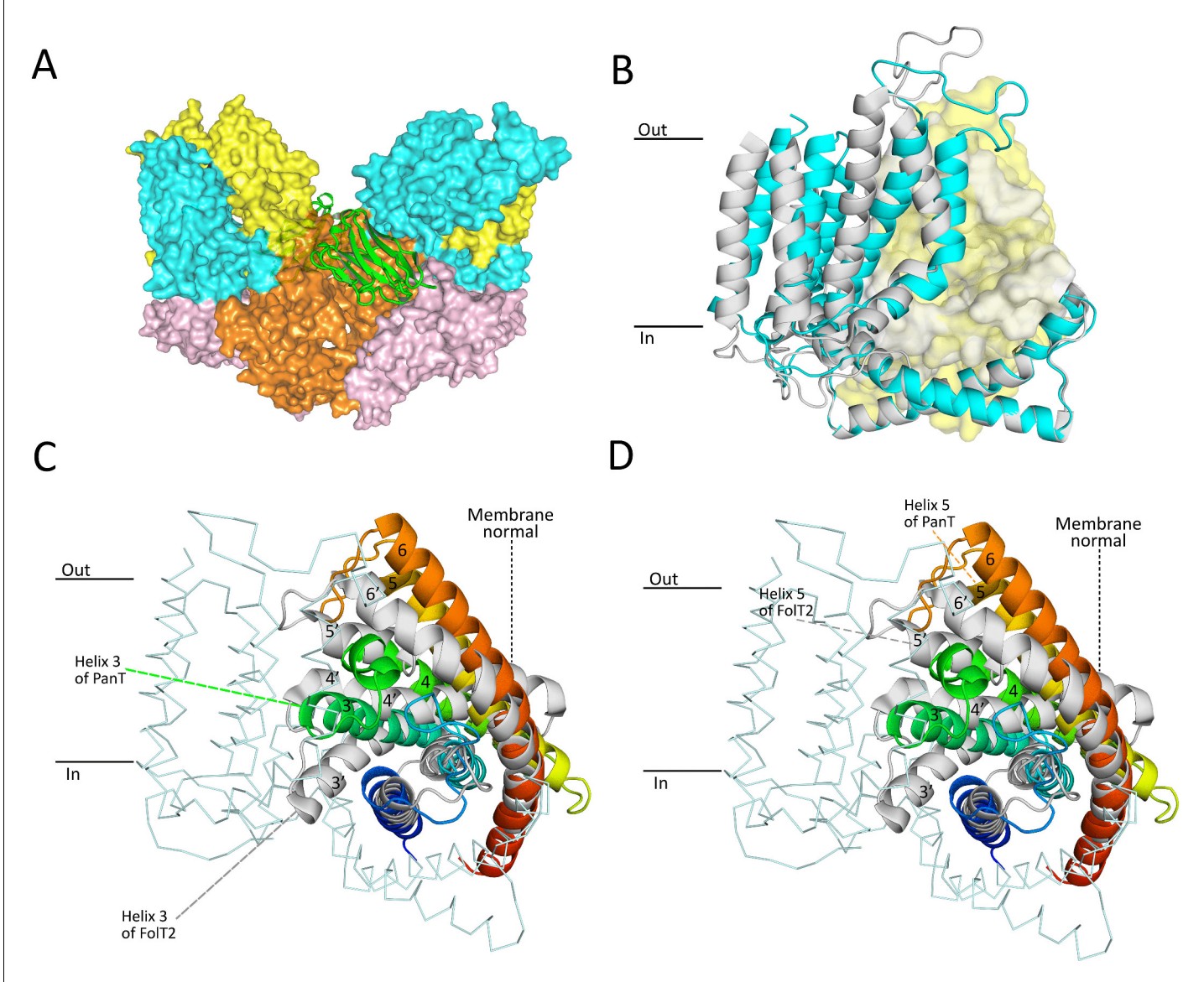

**Figure 6.** Crystal structure of nanobody-bound ECF-PanT. (**A**) Overall structure with two ECF-PanT complexes (in surface representation) bridged by the nanobody (in secondary structure cartoon representation). EcfA in orange, EcfA' in light pink, EcfT in cyan, PanT in yellow, nanobody 81 in green. (**B**) Comparison of the conformations of the membrane domains of EcfT in the structures of ECF-PanT (same colours as in panel A), and ECF-FolT2 (in grey, PDB 5JSZ). The structures were aligned on the ATPase domains which are not shown for clarity, see *Figure 6—figure supplements 3* and *4*. EcfT proteins are shown in secondary structure cartoon representation, the S-components in surface representation. (**C and D**) Comparison of the conformations of the S-components in the structures of ECF-PanT (PanT in rainbow from blue at the N-terminus to red at the C-terminus), and ECF-FolT2 (FolT2 in grey). EcfT from the ECF-PanT structure is shown in ribbon representation. The approximate positions of the membrane boundaries are indicated. Membrane helices are numbered, the ones from FolT2 with an added prime. The differences in membrane orientation of helix 3 (panel **C**) and helix 5 (panel **D**) are indicated by the dashed lines.

The online version of this article includes the following figure supplement(s) for figure 6:

**Figure supplement 1.** Crystal packing of nanobody-bound ECF-PanT.

**Figure supplement 2.** Shifted elution volume of ECF-PanT in gel filtration chromatography, black and blue traces are in the absence and presence of nanobody, respectively.

**Figure supplement 3.** Structural alignment of nanobody-bound ECF-PanT and ECF-FolT2 (PDB 5JSZ).

**Figure supplement 4.** Structural alignment of the ATPase subunits in ECF-PanT (EcfA in orange), (EcfA' in light pink) and ECF-FolT2 (grey, PDB 5JSZ).

**Figure supplement 5.** Structural alignment of the coupling helices in ECF-PanT and ECF-FolT2 (PDB 5JSZ).

**Table 1.** Data collection, phasing and refinement statistics.

*Data collection*

| | |
|---|---|
| Space group | P1 |
| Cell dimensions | |
| a, b, c (Å) | 97.290 110.470 110.500 |
| α, β, γ (°) | 89.00 102.27 102.24 |
| Resolution (Å) | 48.80–2.80 |
| $CC_{1/2}$ | 0.997 (0.195) |
| I/σI | 4.7 (0.77) |
| Completeness (%) | 96.7 (95.3) |
| Multiplicity | 1.76 (1.52) |
| *Refinement* | |
| Resolution (Å) | 48.80–2.80 |
| No. of reflections | 104284 |
| *Rwork/Rfree* | 24.3/27.6 |
| No. of atoms | |
| Protein | 17885 |
| Ligand/ion | 338 |
| Water | - |
| *B*-factors | |
| Protein | 108.6 |
| Ligand/ion | 130.2 |
| Water | - |
| R.m.s. deviations | |
| Bond lengths (Å) | 0.010 |
| Bond angles (°) | 1.286 |

have aided crystal formation. Overall, the structure of ECF-PanT from *L. delbrueckii* is very similar to previously solved structures of ECF-FolT2 from the same organism (*Figure 6—figure supplement 3*), and also to ECF-PanT from and *L. brevis* (*Zhang et al., 2014*), despite only 36% of sequence identity between the PanT subunits of the two organisms. In all these protein complexes, the two ATPase subunits (EcfA and EcfA') are separated from each other, adopting an open conformation, which has been interpreted as a post-hydrolysis state (*Figure 6—figure supplement 4*). The residues in ECF-PanT that interact directly with the nanobody adopt virtually identical conformations as those of ECF-FolT2, which was crystallised without a nanobody chaperone, indicating that the nanobody did not induce an artificial conformation.

Not only the ATPase subunits, but also the coupling helices of EcfT, which mediate the interaction with the NBDs, have almost identical conformations in the structures of ECF-PanT and ECF-FolT2, again indicating that the same functional state was captured (*Figure 6b* and *Figure 6—figure supplement 5*). Within the identical ECF modules of ECF-PanT and ECF-FolT2 from *L. delbrueckii*, the most prominent difference is the relative positioning of the transmembrane-domain of EcfT compared to the coupling helices. In ECF-FolT2 the transmembrane domain is rotated further away from the centre of the complex than in ECF-PanT (*Figure 6b*). Hinging between the two domains has been observed before (*Swier et al., 2016*; *Santos et al., 2018*; *Zhang et al., 2014*) and is likely needed to accommodate structurally different S-components in the complexes (*Santos et al., 2018*).

In contrast to the ECF modules of the ECF-FolT2 and ECF-PanT complexes, the S-components display large structural differences. While on a global level, PanT and FolT2 share conserved six-helix topologies, and both S-components are in the inward-oriented toppled state in complex with the ECF module, there are two prominent differences between the proteins. First, only helices 1 and 2

superimpose well in the ECF-FolT2 and ECF-PanT complexes, with the positions of helices 3–6 deviating substantially by up to 10 Å (when the structures are aligned on the coupling helices [*Figure 6cd*]). In PanT, the latter helices are oriented more perpendicular to the predicted membrane plane (less toppled) than in FolT2. Second, the predicted substrate-binding site in ECF-PanT is located in a largely occluded cavity with a volume of 880 Å$^3$ (*Figure 7a*), whereas in ECF-FolT2 the site is fully accessible from the cytoplasmic side of the membrane. A non-protein patch of electron density was found in the substrate-binding cavity of PanT (*Figure 7bc*). Since pantothenate was not added at any stage during purification and crystallisation the density likely belongs to a molecule from the crystallisation condition, most probably citrate, which was present at a concentration of 70 mM.

## Discussion

Some membrane transporters have evolved to make a use of a single transmembrane pore for many substrates. In the superfamily of ABC transporters, a small number of classical (Type I) importers exist where more than one SBP interacts with the same translocator complex. Different SBPs evolved as gene duplications followed by divergent evolution into two homologous proteins with different substrate specificities or substrate affinities (*Ghimire-Rijal et al., 2014*; *Higgins and Ames, 1981*; *Chen et al., 2010*). In some cases the SBDs fused to TMDs (*Fulyani et al., 2013*; *van der Heide and Poolman, 2002*). Single molecule FRET studies on such fused proteins provided insight into the competing behavior of SBDs interacting with the shared translocator (*Gouridis et al., 2015*). In all these cases, multiple substrate binding proteins interact with the shared transmembrane translocator, and competition of the SBPs or SBDs for the shared part of the transporter resembles what is observed in ECF transporters, but exceptionally, the latter transporters use integral membrane binding proteins (S-components) instead of SBPs, which compete with each other for the ECF module within the lipid bilayer environment.

To study the dynamic interaction and competition in ECF transporters in vitro, we had to find two group II ECF transporters from the same organism that were functional upon purification and reconstitution into the proteoliposomes. The transporters ECF-FolT2 and ECF-PanT from *L. delbrueckii* fulfilled these criteria. While ECF-FolT2 had been shown to transport folate in a reconstituted system before, we here show for the first time that purified and reconstituted ECF-PanT catalyses pantothenate transport. Co-reconstitution experiments showed that the S-components PanT and FolT2, which share only 21.5% sequence identity, dynamically associate with and dissociate from the common ECF module. Dynamic interaction in the lipid bilayer explains the observed transport of folate when an incomplete or inactive folate transporter (FolT2 alone, or FolT2 in complex with a mutated ECF

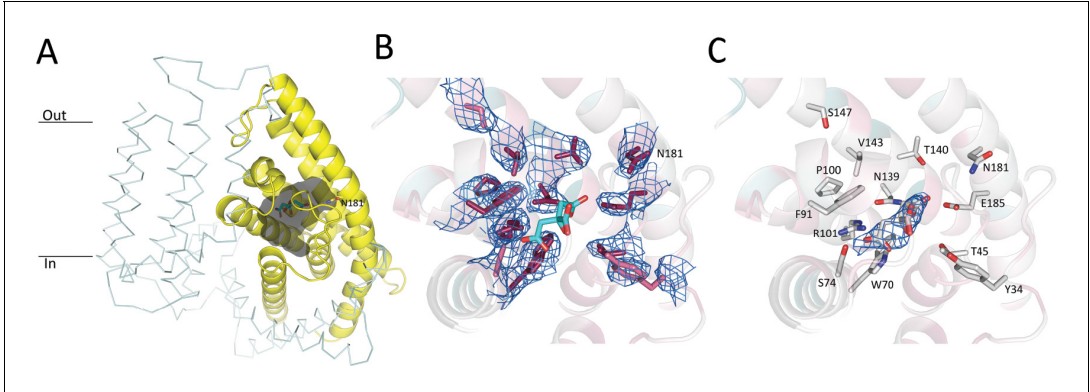

**Figure 7.** Pantothenate binding pocket in ECF-PanT. (**A**) Binding pocket (grey) of the PanT (yellow, secondary structure cartoon) EcfT is shown in ribbon representation, ATPases are not shown for clarity. The approximate positions of the membrane boundaries are indicated. The modelled citrate molecule is shown in stick representation. (**B**) Electron 2Fo-Fc density contoured at 1.0 σ for conserved residues in the binding pocket. Colouring of the side chains according to conservation as calculated by the Consurf server (*Landau et al., 2005*). Dark purple indicates highly conserved residues. The modelled citrate molecule is shown in stick representation with carbon atoms in cyan and oxygen atoms in red. (**C**) Electron 2Fo-Fc density contoured at 1.0 σ for modelled citrate molecule in the binding pocket.

module, respectively) was reconstituted together with fully active ECF-PanT complexes, and vice versa (*Figure 3,4*).The data strongly suggest that association and dissociation of S-components is an essential step in the transport mechanism in group II ECF transporters.

Remarkably, the rates of both folate and pantothenate transport were consistently higher in liposomes containing both ECF-PanT and FolT2 than in liposomes containing only ECF-FolT2 or ECF-PanT, respectively (Compare *Figure 1* with *Figure 3a* and *Figure 3—figure supplement 1*). Although this difference may originate from the reconstitution procedure, for instance the reconstitution efficiency might be affected on the total amount of purified protein used, it is also possible that it reflects a mechanistic feature of ECF transporters. The excess of FolT2 molecules in the co-reconstituted system might cause bilayer imperfections, which facilitate toppling (*Faustino et al., 2020*) thus leading to increased transport rates. Further experimental work is needed to test this speculative explanation.

We showed that FolT2 and PanT, when bound to their respective transported substrates, compete for the same ECF module (*Figure 5*), thereby nicely recapitulating previous in vivo work (*Henderson et al., 1979a*; *Majsnerowska et al., 2015*), from which it was deduced that substrate-bound S-components compete more efficiently for the ECF module than those without substrate. Also, the observation that the extent of inhibition of [³H]folate uptake by pantothenate and vice versa, depends on the relative amounts of the S-components in the proteoliposomes (*Figure 5B*; *Figure 5D*), is fully consistent with published in vivo observations (*Henderson et al., 1979a*; *Majsnerowska et al., 2015*).

Since the number of substrate molecules that was transported into the lumen of the liposomes was higher than the number of ECF complexes present in the liposomal membranes, multiple turnovers per transporter complex occurred in the experiments presented in *Figure 5a–c*. Therefore, the observed competition is not caused by a half cycle leading to transporters in a dead-end conformation. Only for the folate transport experiments presented in *Figure 5d*, the data could suggest that less than one folate molecule is transported per ECF transporter. However, if we take into account that the proteins can reconstitute in two orientations in the membrane (*Swier et al., 2016*), and that most likely some activity was lost during the purification and reconstitution procedure, it is reasonable to assume that multiple turnovers also took place in this case. This conclusion is further supported by the notion that multiple (unlabelled) pantothenate molecules per protein complex must have been transported in the same experiment, as deduced from the experiment presented in *Figure 5c* where more than one turnover of the pantothenate transporter was observed when radiolabelled substrate was in an identical liposome preparation as used for *Figure 5d*.

In the co-reconstituted system, we observed that increasing amounts of unlabelled folate compete with [³H]pantothenate uptake in a dose-dependent manner in the range of 0.5–50 µM folate. Surprisingly, this concentration regime is ~3 orders of magnitude higher than the $K_D$ for folate binding to FolT2 (*Swier et al., 2016*) and the $K_M$ for folate transport (*Figure 2*). To explain this discrepancy, we hypothesise that the binding of folate must be much slower than the binding of pantothenate, albeit not the rate-limiting step, as the $V_{max}$ values for substrate transport by both transporters are similar (*Figure 2*). The presumed slow substrate association does not lead to poor affinity of FolT2 for folate (the $K_D$ value for folate binding to FolT2 is in the nanomolar range), but is most likely caused by a slow conformational change in the protein, either preceding binding (conformational selection) or following an initial low affinity association (induced fit). In case of [³H]folate uptake, inhibition by pantothenate was also observed, but did not show dose-dependence in the 0.5–50 µM range, which suggests that pantothenate binding is faster compared to folate binding. Alternative explanations for the discrepancy, such as differences in the dissociation rate of PanT and FolT2 from the ECF module, affected additionally by the presence of substrate, are also possible but require more assumptions. For instance, to explain the folate dependence of competition (*Figure 5a and c*) by slower dissociation of FolT2 than PanT from the ECF module, it is necessary to postulate that a state must exist in which the full complex (ECF module and FolT2) has an outward facing substrate-binding site of low affinity. There is currently no structural evidence for such a state.

In contrast, the explanation based on slow binding of folate to solitary FolT2 can be explained from a structural viewpoint, because crystal structures are available of the full complex ECF-FolT2 with the S-component in the apo state, and the S-component FolT1 alone with folate-bound (*Swier et al., 2016*). The conformation of the full ECF-FolT2 complex was interpreted as an inward-open, post-release state, from which the transported substrate has been delivered in the cytoplasm,

whereas the structure of the solitary S-component with bound folate was interpreted as a state in which substrate had been captured from the environment, before association with ECF module. Substantial structural differences between the folate-bound and *apo* S-component may explain the slow binding of folate, particularly in the conformations of loops L1 and L3 that strongly affect the binding site geometry (*Swier et al., 2016*). To find a potential structural explanation for the apparent faster binding of pantothenate, we solved a crystal structure of ECF-PanT. Because the conformations of the ECF modules in ECF-PanT and ECF-FolT2 are very similar, we also interpret the ECF-PanT structure as a post-release state. Although we do not have a substrate-bound structure of solitary PanT for comparison, the analysis of the ECF-PanT structure provides clues about possible differences in kinetics of folate and pantothenate binding. First, the binding pocket in PanT is more occluded, with loops L1 and L3 not being splayed out as far as in the ECF-FolT2 structure (*Figure 6a*). Therefore, smaller conformational changes are expected upon pantothenate binding to PanT than folate binding to FolT2. Second, in contrast to what was observed in ECF-FolT2, residues in ECF-PanT that have been shown by mutational analysis to be important for binding of pantothenate (R101(95), N139(131), W69(64), residue numbers of PanT from *L. brevis* in parentheses) (*Zhang et al., 2014*) all point towards the centre of the binding pocket (*Figure 6bc*). This binding site geometry of the apo state again suggests that only minor rearrangements are needed for pantothenate binding. It may be argued that the structure of ECF-PanT from *L. delbrueckii* presented here does not represent a true apo state, as a patch of electron density was found in the pocket, which likely results from a bound citrate molecule form the crystallisation condition. However, the binding site organisation in PanT is identical to that of a previously published structure of ECF-PanT from *L. brevis*, which represents a true apo state (*Zhang et al., 2014*), and therefore citrate molecule does not appear to affect the geometry of binding site of the apo state.

Possibly, the presumed fast binding of pantothenate to PanT resembles that of thiamin binding to the S-component ThiT, where pre-steady-state fluorescence experiments showed rapid association kinetics (*Majsnerowska et al., 2013*). In the case of ThiT, only a structure of the thiamin-bound S-component is available, and not a structure of the apo-full complex, which again makes a complete structural comparison as for ECF-FolT2 impossible. It is noteworthy that previously, the structures of FolT1 and ECF-FolT2 were interpreted as being consistent with fast binding kinetics, albeit without any experimental kinetics data (*Swier et al., 2016*). The work presented here shows that care needs to be taken when extracting kinetic behavior from static structures.

In conclusion, the relatively simple reconstituted systems that we have used here, is sufficient to reproduce the competition between S-components for the same ECF module as observed in vivo, but in addition, more intricate kinetic differences between transport of folate and pantothenate also became apparent. Combining the kinetic measurements with structural analysis yielded a potential mechanistic explanation for the differences in association rates. More generally, because dissociation and association of S-components are essential steps in the transport cycle, and multiple S-components interact with the same ECF module, ECF transporters may serve as a model system for studying membrane protein interaction in the lipid bilayers.

# Materials and methods

**Key resources table**

| Reagent type (species) or resource | Designation | Source or reference | Identifiers | Additional information |
|---|---|---|---|---|
| Gene (*Lactobacillus delbrueckii* subsp. *bulgaricus*) | *panT* | GenBank: CP002341.1 | LDBND_0406 | |
| Strain, strain background (*Escherichia coli*) | MC1061 | *Casadaban and Cohen, 1980* | | |

*Continued on next page*

*Continued*

| Reagent type (species) or resource | Designation | Source or reference | Identifiers | Additional information |
|---|---|---|---|---|
| Strain, strain background (*E. coli*) | WK6 | ATCC 47078 | | |
| Biological sample (*L. delbrueckii*) | *L. delbrueckii* subsp. bulgaricus genomic DNA | DSMZ | DSM 20081 | |
| Recombinant DNA reagent | pBAD24_PanT | This paper | | Expression plasmids for PanT in *E. coli*. Plasmid can be provided upon reasonable request. |
| Recombinant DNA reagent | p2BAD_ECF_panT | This paper | | Expression plasmid for ECF-PanT in *E. coli*. Plasmid can be provided upon reasonable request. |
| Recombinant DNA reagent | pMESy4 | GenBank KF415192 | | |
| Strain, strain background (*E. coli*) | TG1 | https://ecoliwiki.org/colipedia/index.php/Category:Strain:TG1 | | |
| Reagent | pantothenic acid, D-[2,3-$^3$H] sodium salt | American Radiolabelled Chemicals | | |
| Reagent | folic acid [3,5,7,9-$^3$H] sodium salt | American Radiolabelled Chemicals | | |
| Other | ECF-PanT coordinate file and structure factors | this paper | accession number PDB ID code 6ZG3 | Crystal structure of ECF-PanT |

## Mutagenesis

Mutations in EcfA and EcfA' of ECF-FolT2 and ECF-PanT were introduced by two consecutive rounds of QuikChange mutagenesis with primers listed below.

| Primer name (mutation) | Primer sequence (5'→3') |
|---|---|
| Fw EcfA E177Q Ldb (E169Q in wild type) | CATCATCCTGGATCAGTCGACCTCCATG |
| Rev EcfA E177Q Ldb (E169Q in wild type) | CATGGAGGTCGACTGATCCAGGATGATG |
| Fw EcfA' E171Q Ldb | TGTTTAGATCAGCCGGCAGCTGG |
| Rev EcfA' E171Q Ldb | CCAGCTGCCGGCTGATCTAAACA |

## Expression and membrane vesicles preparation

The genes encoding ECF-PanT and ECF-FolT2 from *L. delbrueckii* subsp. *bulgaricus* (LDB_RS01805, *ecfA*; LDB_RS01810, *ecfA'*; LDB_RS01815, *ecfT*; LDB_RS01970, *panT*; LDB_RS07030, *folT2*) were cloned in p2BAD vectors and transformed into Ca$^{2+}$-competent cells of the *Escherichia coli* strain MC1061 as described before (*Swier et al., 2016*; *Birkner et al., 2012*). The ECF module operon (10xHis-TEV-*ecfAA'T*) was cloned downstream the first arabinose promoter and the gene encoding PanT or FolT2 (*panT*-Strep or *folT2*-Strep, respectively) downstream of the second arabinose

promoter. The expression from p2BAD plasmids was performed in 2 L of LB Miller Broth containing 0.1 mg/mL ampicillin in a 5 L flask. The *E. coli* culture was grown at 37°C with continuous shaking at 200 rpm. At $OD_{600}$ between 0.6 and 0.8, the expression from p2BAD plasmids was induced with 0.1 mg/mL of L-arabinose and the temperature was reduced to 25°C for three hours. Cells were harvested by centrifugation for 15 min at 6268 x g, 4°C.

Solitary S-components from *L. delbrueckii* were engineered with N-terminal $His_{10}$ tag and cloned in pNZ8048 plasmids with the gene coding for either PanT or FolT2 protein downstream of the nisin promoter (*Erkens et al., 2011*; *Swier et al., 2016*; *Duurkens et al., 2007*). For expression, the constructed vectors were transformed into *Lactococcus lactis* NZ9000 cells. This expression was performed semi-anaerobically in a 1 L bottle with M17 media (Difco), 5 µg/mL chloramphenicol, and 2.0% (w/v) glucose at 30°C. Overexpression of the solitary S-component was induced at $OD_{600}$ around 0.8 with 0.1% (v/v) of the supernatant of a nisin A producing strain. The cells were harvested by centrifugation (15 min, 6268 × g at 4°C) after the 3-h expression.

Membrane vesicles were prepared as described previously (*ter Beek et al., 2011*). Briefly, harvested cells were diluted to an $OD_{600}$ of around 100 with potassium phosphate buffer pH 7.5 and supplemented with 1 mM $MgSO_4$ and DNase (~50 µg/mL). The cells were broken in a Constant cell Disruption System (Constant Systems Ltd) in the presence of 1 mM PMSF and 5 mM EDTA. For *E. coli* cells one passage at 25 kPsi and for *L.lactis* cells two passages at 39 kPsi were performed. Unbroken cell debris was separated by low-speed centrifugation (15 min, 27352 × g at 4°C). Subsequently, the membranes were concentrated by ultracentrifugation (120 min, 193,727 × g at 4°C), homogenised in 50 mM potassium phosphate buffer pH 7.5, flash-frozen and stored at −80°C.

## Protein purification and reconstitution into proteoliposomes

For the whole complex ECF-transporter purification, membrane vesicles were thawed and incubated for 1 hr with 1% (w/v) *n*-dodecyl-β-D-maltopyranoside (DDM, Anatrace) in buffer containing 50 mM potassium phosphate pH 7.5, 300 mM NaCl, and 10% (v/v) glycerol. Non-solubilised membrane fragments were removed by centrifugation (35 min, 286286 × g at 4°C). The solubilised protein solution was mixed with nickel-Sepharose resin equilibrated with solubilisation buffer and incubated for 1 hr with gentle rocking at 4°C. Proteins not bound to the resin were drained and subsequently washed away with 20 column volumes of 50 mM potassium phosphate buffer pH 7.5 supplemented with 300 mM NaCl, 50 mM imidazole pH 7.5% and 0.05% (w/v) DDM. The protein was eluted from the Ni-Sepharose column in three steps (fraction volumes 350 µL, 750 µL and 700 µL, respectively) in 50 mM potassium phosphate buffer pH 7.5 supplemented with 300 mM NaCl, 500 mM imidazole pH 7.5% and 0.05% (w/v) DDM. The second fraction, the one with the highest protein content, was supplemented with 1 mM Na-EDTA and further purified by size-exclusion chromatography on a Sephadex200 10/300 column (GE Healthcare) using 50 mM potassium phosphate buffer pH 7.5 supplemented with 150 mM NaCl and 0.05% (w/v) DDM as eluent. Peak protein fractions after the size-exclusion chromatography were used for the protein reconstitution into liposomes according to a previously described method (*Swier et al., 2016*; *ter Beek et al., 2011*; *Geertsma et al., 2008*). Liposomes were composed of *E. coli* polar lipids supplemented with 1/3 (w/w) egg phosphatidylcholine with final protein-to-lipid ratio in liposomes 1:1000 (w/w).

Solitary FolT2 was purified using the same approach as for the complete ECF transporter complexes, but with slight modifications. In all buffers NaCl was replaced with KCl. Membrane vesicles were solubilised with 1% (w/v) DDM, but thereafter 0.38% (w/v) *n*-nonyl-β-D-glucopyranoside (NG, Anatrace) was used instead of DDM to purify the solubilised FolT2 by Ni-Sepharose and size-exclusion chromatography. Solitary S-components were reconstituted into the detergent-destabilised liposomes with protein-to-lipid ratios 1:250 or 1:1000 (w/w).

The co-reconstitution of multiple proteins was performed in the same manner as for individual reconstitution, always maintaining each protein-to-lipid ratio separately.

## Transport assays

Transport assays using radiolabelled substrates were performed as described previously with some modifications (*Swier et al., 2016*). Briefly, inclusion of 10 mM (unless otherwise indicated) $Mg^{2+}$-ATP or $Mg^{2+}$-ADP into proteoliposomes was achieved by three consecutive cycles of flash-freezing in liquid nitrogen and thawing at room temperature, followed by 11 passages of extrusion through a

polycarbonate filter (Avestin) with the pore size 400 nm. The remaining external nucleotides were removed by ~15-fold dilution of the proteoliposomes in 50 mM potassium buffer (final volume of 7 mL) followed by centrifugation (45 min, 286286 × g at 4°C). Subsequently, the proteoliposomes were resuspended in 50 mM potassium phosphate pH 7.5 to a protein concentration of 1.25–2.5 μg/mL. Substrate uptake assays were performed at 30°C with stirring and initiated by adding the transported substrate (mixture of 5 nM radiolabelled and 95 nM non-radiolabelled substrate). For folate transport assays folic acid [3,5,7,9 -$^3$H] sodium salt (American Radiolabelled Chemicals) and for pantothenate transport assays pantothenic acid, D-[2,3-$^3$H] sodium salt (American Radiolabelled Chemicals) were used. At given time intervals, 200 μL of the reaction mixture was withdrawn and diluted in ice cold 50 mM potassium phosphate pH 7.5, followed by immediate collection of the proteoliposomes by filtration over pre-wetted cellulose nitrate filters. Subsequently, filters were washed with 2 mL of 50 mM potassium buffer, dried for at least 1 hr at 80°C and dissolved in 5 mL of Filter Count scintillation liquid (Perkin Elmer). The radioactivity trapped inside the proteoliposomes was determined with a Perkin Elmer Tri-carb 2800TR Scintillation counter.

## Expression and purification of nanobodies

For nanobody generation, a llama (*Lama glama*) was immunised as in reference (*Pardon et al., 2014*) with 800 μg ECF-PanT which had been reconstituted in liposomes consisting of an *E. coli* polar lipids-phosphatidylcholine (3:1 w/w ratio) mixture as descibed above, using a protein-to-lipid ratio of 1:125 (w/w). A phage display library of nanobodies modified by introducing a C-terminal His$_6$ and EPEA tags via PCR was prepared from peripheral blood lymphocytes, and the open reading frames of the nanobodies were cloned as *Sap*I digested fragments in a Golden Gate variant of pMESy4 (GenBank KF415192) and subsequently transformed to *E. coli* TG1 to establish a library of 7E9 independent Nb clones. The phage display selections were performed using either solid-phase immobilised ECF-PanT proteoliposomes or was captured on anti-Strep-tag mAbs coated Maxisorp plates. 21 nanobody families were identified that specifically had bound the ECF-PanT protein, one of which included the nanobody selected for crystallisation and structure determination (nanobody CA14381 or Nb81).

The nanobodies were expressed in the periplasm of *E. coli* strain WK6 (su-), following methods described previously (*Pardon et al., 2014*). Briefly, 1 L cultures in Terrific Broth were grown to an OD$_{600}$ of 1.0–1.2 and induced with 1 mM isopropyl-b-D-thiogalactoside (IPTG). Cells were harvested after overnight growth at 25°C, and periplasmic extract prepared using TES (Tris EDTA Sucrose) buffer. Nanobodies were purified from the periplasmic extract by Ni-Sepharose column. The nanobody was eluted from the Ni-Sepharose column using an elution buffer containing 50 mM potassium phosphate pH 7.5, 150 mM NaCl and 300 mM imidazole. Subsequently, the imidazole in the nanobody fraction was removed by using desalting column (GE Healthcare).

## Co-Purification of ECF-PanT with the nanobody

The purified nanobody was mixed with ECF-PanT that had been purified by Ni-Sepharose chromatography as described above, and the mixture was applied to a gel filtration column (Superdex 200 10/300, GE Healthcare), using a buffer containing 50 mM Tris HCl pH 7.5, 150 mM NaCl and 0.05% (w/v) DDM, as described above. The fractions containing the purified complex were directly concentrated to 5-6 mg/mL by the use a concentrating device (Vivaspin 500, Sartorius, molecular weight cut off 100 kDa) and used for crystallisation.

## Crystallisation

Initial crystallisation conditions were screened using 5 mg/mL of ECF-PanT-Nanobody 81 complex mixed with 5 mM MgATPγS, at 5°C using the MemGold and MemGold2 HT-96 solutions (Molecular Dimensions, UK) in a sitting-drop setup with a Mosquito robot (TTP Labtech, UK) with drop ratios of 100 nL protein and 100 nL precipitating solution. The crystals were found in the G9 condition (70 mM sodium citrate, pH 4.5 and 22% (v/v) PEG300) of the MemGold2 screen. Using this condition, the crystallisation was set up in a bigger volume (2 μL protein and 2 μL precipitating solution) in 24-well hanging drop vapor diffusion plates combined with a streak seeding technique. Crystallisation plates were incubated at 5°C and rod-shaped crystals appeared within 2 weeks. Crystals were

harvested from the drops, cryo-protected with a solution containing 70 mM sodium citrate, pH 4.5 and 40% (v/v) PEG300, followed by flash-freezing in liquid nitrogen.

## Data collection and structure determination

Diffraction data for the Ecf PanT-Nanobody crystals were collected at 100 K at Diamond Light Source beamline I24 with the highest diffraction limit of 2.8 Å resolution. The crystal belongs to space group P1 (unit cell parameters: a = 97.290 Å, b = 110.470 Å, c = 110.500 Å, $\alpha$=89.00°, $\beta$=102.27°, $\gamma$=102.24°). Data sets were indexed, integrated and scaled using the programs XDS (*Kabsch, 2010*) and molecular replacement was carried out with PHASER MR (*Kabsch, 2010*). Data collection and refinement statistics are summarised in *Table 1*. The *apo* ECF-FolT2 structure of *L. delbruckii* (PDB code 5JSZ) (*Swier et al., 2016*) was used as a search model for the EcfA, EcfA' and EcfT subunits. However, attempts to use the published PanT structure of *L. brevis* (PDB code 4RFS) (*Zhang et al., 2014*) to find the position of the PanT subunit failed. To overcome this problem and to reduce possible bias, Rosetta-based MR was used (*DiMaio et al., 2011*). The refinement was performed with Phenix refine (*Adams et al., 2010*), with the model building done with COOT (*Emsley et al., 2010*).

## Acknowledgements

We acknowledge the support and the use of resources of Instruct-ERIC (PID 6357), part of the European Strategy Forum on Research Infrastructures (ESFRI), Instruct-ULTRA (EU H2020 Grant 731005) and the Research Foundation - Flanders (FWO) for their support to the Nanobody discovery. We further acknowledge Nele Buys, Eva Beke, Katleen Willibal and Allison Lundqvist for the technical assistance during Nanobody discovery and Instruct-ERIC Nanobody workshop. We thank Josy ter Beek for cloning the genes encoding ECF-PanT. IS received a scholarship from the Indonesia Endowment Fund for Education (LPDPLembaga Pengelolaan Dana Pendidikan, Departemen Keuangan, Republik Indonesia). DJS acknowledges support from NWO (TOP grant 714.018.003).

## Additional information

### Funding

| Funder | Grant reference number | Author |
| --- | --- | --- |
| Nederlandse Organisatie voor Wetenschappelijk Onderzoek | TOP grant 714.018.003 | Dirk J Slotboom |
| Fonds Wetenschappelijk Onderzoek | | Els Pardon<br>Jan Steyaert |
| Horizon 2020 | Grant 731005 | Jan Steyaert |
| Instruct-ERIC | PID 6357 | Albert Guskov<br>Dirk J Slotboom |

The funders had no role in study design, data collection and interpretation, or the decision to submit the work for publication.

### Author contributions

Inda Setyawati, Formal analysis, Investigation, Visualization, Writing - original draft, Writing - review and editing; Weronika K Stanek, Conceptualization, Formal analysis, Investigation, Methodology, Writing - original draft, Writing - review and editing; Maria Majsnerowska, Conceptualization, Formal analysis, Investigation; Lotteke J Y M Swier, Formal analysis, Investigation; Els Pardon, Formal analysis, Investigation, Methodology, Writing - review and editing; Jan Steyaert, Methodology; Albert Guskov, Formal analysis, Supervision, Validation, Visualization, Writing - review and editing; Dirk J Slotboom, Conceptualization, Formal analysis, Supervision, Funding acquisition, Writing - original draft, Writing - review and editing

## Author ORCIDs
Weronika K Stanek  https://orcid.org/0000-0001-7846-3927
Els Pardon  http://orcid.org/0000-0002-2466-0172
Jan Steyaert  http://orcid.org/0000-0002-3825-874X
Albert Guskov  http://orcid.org/0000-0003-2340-2216
Dirk J Slotboom  https://orcid.org/0000-0002-5804-9689

## Decision letter and Author response
Decision letter https://doi.org/10.7554/eLife.64389.sa1
Author response https://doi.org/10.7554/eLife.64389.sa2

# Additional files
## Supplementary files
• Transparent reporting form

## Data availability
All data generated or analysed during this study are included in the manuscript and supporting files. Source data files have been provided for Figures 1–5.

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
