## [Decision Letter]

**Acceptance summary:**

Previous in vivo experiments have suggested that different substrate-binding S-components of ECF-II ABC transporters share a common ECF module, broadening the substrate specificity of the transport system. The definitive proof, however, has been long missing. This paper provides that evidence, using a reconstituted system to clearly establish that the ECF transport module of the ECF-II ABC transporters from *Lactobacillusdelbrueckii* can mediate transport of either folate or pantothenate, depending on its interactions with the substrate-binding S-component. This paper is also notable because it establishes a reconstituted system in which integral membrane proteins dynamically associate and dissociate in the lipid bilayer, a technical achievement that will support future work to understand the physical and chemical factors that contribute to membrane protein interactions.

**Decision letter after peer review:**

Thank you for submitting your article "in vitro reconstitution of dynamically interacting integral membrane subunits of Energy-Coupling Factor transporters" for consideration by *eLife*. Your article has been reviewed by three peer reviewers, including Randy B Stockbridge as the Reviewing Editor and Reviewer #1, and the evaluation has been overseen by Kenton Swartz as the Senior Editor. The following individual involved in review of your submission has agreed to reveal their identity: Da-Neng Wang (Reviewer #2).

The reviewers have discussed the reviews with one another and the Reviewing Editor has drafted this decision to help you prepare a revised submission.

Summary:

The ECF ABC transporters are a subclass of ABC transporters that use integral membrane proteins to bind and deliver substrate. Previous in vivo experiments have suggested that various substrate-binding S-components of ECF-II ABC transporters in a bacterial species share a common ECF module, broadening the substrate specificity of the transport system. The definitive proof, however, has been long missing. In this paper, Setyawati and colleagues have used a reconstituted system to provide strong evidence for this theory, clearly establishing that the ECF module from *Lactobacillusdekbrueckii* can mediate transport of either folate or pantothenate, depending on the S-component binding domain. In addition, the authors solve the structure of ECF-PanT and find interesting differences between it and the previously determined structure of ECF-FolT2. This is a thoughtful and comprehensive set of experiments that represent a substantial technical achievement. Overall, the work provides a major advance in the understanding of this type of intriguing ABC transporters. The reviewers have some suggestions for improving the interpretation and presentation of the experiments.

Essential revisions:

1) The experiments themselves are quite complex, varying many components. It would be very helpful to have some kind of visual guide to what experiment is being done for each graph; perhaps a small cartoon of which proteins and substrates are present. In addition, adding color to the graphs, consistently across the work, could aid in guiding the readers' thinking.

2) The speculation regarding the effects of high concentrations of S-factor facilitating toppling (subsection “Exchange of S-components”) may go a bit far, given that it seems like only one of many possible explanations for the differences in rates. The reviewers suggest that this topic should be moved to and elaborated on in the Discussion.

3) In the competition experiment with co-reconstituted ECF-PanT and FolT2, the components are assumed to insert in either right-side- and inside-out orientation (subsection “Competition for a shared ECF module”, and Figure 5), which is reasonable. However, the directionality of proteins in other reconstituted systems is not discussed. As a result, it is unclear whether this would affect the interpretation of the experiment. For example, were all proteins assumed to be right-side-out in the experiments for measuring the Vmax (Figure 2)? Were some of the ECTs inside-out and therefore "invisible"? How would that affect the V_max_ and K_m_ calculations?

4) Do the authors have evidence regarding whether there is more than one turnover event per liposome? This seems like an important point for thinking about the differences between pantothenate and folate inhibition. Could the authors comment on whether the dynamics of SBD exchange might contribute to these differences (for example, differences in the dissociation rates of the apo PanT and FolT)?

---

## [Author Response]

Essential revisions:1) The experiments themselves are quite complex, varying many components. It would be very helpful to have some kind of visual guide to what experiment is being done for each graph; perhaps a small cartoon of which proteins and substrates are present. In addition, adding color to the graphs, consistently across the work, could aid in guiding the readers' thinking.

We fully agree with the reviewers, and have now added small colored cartoons in each of the panels of Figures 1, 3, 4, 5 to show the proteins present, and colored the symbols in the graphs according to the substrate transported (yellow for pantothenate, green for folate).

2) The speculation regarding the effects of high concentrations of S-factor facilitating toppling (subsection “Exchange of S-components”) may go a bit far, given that it seems like only one of many possible explanations for the differences in rates. The reviewers suggest that this topic should be moved to and elaborated on in the Discussion.

We have now moved the speculation to the Discussion section, where we now clearly state that there may be alternative explanations and that we are speculating:

“Remarkably, the rates of both folate and pantothenate transport were consistently higher in liposomes containing both ECF-PanT and FolT2 than in liposomes containing only ECF-FolT2 or ECF-PanT, respectively (Compare Figure 1 with Figure 3A and Figure 3—figure supplement 1). […] Further experimental work is needed to test this speculative explanation.”

3) In the competition experiment with co-reconstituted ECF-PanT and FolT2, the components are assumed to insert in either right-side- and inside-out orientation (subsection “Competition for a shared ECF module”, and Figure 5), which is reasonable. However, the directionality of proteins in other reconstituted systems is not discussed. As a result, it is unclear whether this would affect the interpretation of the experiment. For example, were all proteins assumed to be right-side-out in the experiments for measuring the Vmax (Figure 2)? Were some of the ECTs inside-out and therefore "invisible"? How would that affect the V_max_ and K_m_ calculations?

We have moved the remarks on the orientation to the section where the K_m_ and V_max_ values are presented:

“Moreover, both protein complexes can orient either in the right-side-out or inside-out orientation in the liposomal membrane (Swier, Guskov and Slotboom, 2016). […] Therefore, while a mixed orientation does not affect the K_m_, but the apparent V_max_ values are likely underestimations.”

4) Do the authors have evidence regarding whether there is more than one turnover event per liposome? This seems like an important point for thinking about the differences between pantothenate and folate inhibition. Could the authors comment on whether the dynamics of SBD exchange might contribute to these differences (for example, differences in the dissociation rates of the apo PanT and FolT)?

The question about multiple turnovers is an important one to address more explicitly. Indeed, multiple turnover take place in the experiments (at least three to four in most experiments). We now added a discussion on this point:

“Since the number of substrate molecules that was transported into the lumen of the liposomes was higher than the number of ECF complexes present in the liposomal membranes, multiple turnovers per transporter complex occurred in the experiments presented in Figure 5A-C. […] This conclusion is further supported by the notion that multiple (unlabelled) pantothenate molecules per protein complex must have been transported in the same experiment, as deduced from the experiment presented in Figure 5C where more than one turnover of the pantothenate transporter was observed when radiolabelled substrate was in an identical liposome preparation as used for Figure 5D.”

We have added a brief discussion on the possibility that alternative steps could explain the differences between pantothenate and folate inhibition:

“Alternative explanations for the discrepancy, such as differences in the dissociation rate of PanT and FolT2 from the ECF module, affected additionally by the presence of substrate, are also possible but require more assumptions. For instance, to explain the folate dependence of competition (Figure 5A and C) by slower dissociation of FolT2 than PanT form the ECF module, it is necessary to postulate that a state must exist in which the full complex (ECF module and FolT2) has an outward facing substrate binding site of low affinity. There is currently no structural evidence for such a state.”